# On the Minimal Supervision for Training Any Binary Classifier from Only Unlabeled Data

**Nan Lu**[1,2]    **Gang Niu**[2]    **Aditya Krishna Menon**[3]    **Masashi Sugiyama**[2,1]

[1]The University of Tokyo, Tokyo 113-0033, Japan
[2]RIKEN, Tokyo 103-0027, Japan
[3]Google, New York, NY 10018, USA

`lu@ms.k.u-tokyo.ac.jp, gang.niu@riken.jp,`
`adityakmenon@google.com, sugi@k.u-tokyo.ac.jp`

## Abstract

*Empirical risk minimization* (ERM), with proper loss function and regularization, is the common practice of supervised classification. In this paper, we study training *arbitrary* (from linear to deep) *binary classifier* from *only unlabeled* (U) *data* by ERM. We prove that it is impossible to estimate the risk of an arbitrary binary classifier in an *unbiased* manner given a single set of U data, but it becomes possible given two sets of U data with *different class priors*. These two facts answer a fundamental question—what the minimal supervision is for training any binary classifier from only U data. Following these findings, we propose an ERM-based learning method from two sets of U data, and then prove it is *consistent*. Experiments demonstrate the proposed method could train deep models and outperform state-of-the-art methods for learning from two sets of U data.

## 1 Introduction

With some properly chosen loss function (e.g., Bartlett et al., 2006; Tewari & Bartlett, 2007; Reid & Williamson, 2010) and regularization (e.g., Tikhonov, 1943; Srivastava et al., 2014), *empirical risk minimization* (ERM) is the common practice of supervised classification (Vapnik, 1998). Actually, ERM is used in not only supervised learning but also *weakly-supervised learning*. For example, in *semi-supervised learning* (Chapelle et al., 2006), we have very limited labeled (L) data and a lot of unlabeled (U) data, where L data share the same form with supervised learning. Thus, it is easy to estimate the risk from only L data in order to carry out ERM, and U data are needed exclusively in regularization (including but not limited to Grandvalet & Bengio, 2004; Belkin et al., 2006; Mann & McCallum, 2007; Niu et al., 2013; Miyato et al., 2016; Laine & Aila, 2017; Tarvainen & Valpola, 2017; Luo et al., 2018; Kamnitsas et al., 2018).

Nevertheless, L data may differ from supervised learning in not only the amount but also the form. For instance, in *positive-unlabeled learning* (Elkan & Noto, 2008; Ward et al., 2009), all L data are from the positive class, and due to the lack of L data from the negative class it becomes impossible to estimate the risk from only L data. To this end, a two-step approach to ERM has been considered (du Plessis et al., 2014; 2015; Niu et al., 2016; Kiryo et al., 2017). Firstly, the risk is rewritten into an equivalent expression, such that it just involves the same distributions from which L and U data are sampled—this step leads to certain risk estimators. Secondly, the risk is estimated from both L and U data, and the resulted empirical training risk is minimized (e.g. by Robbins & Monro, 1951; Kingma & Ba, 2015). In this two-step approach, U data are needed absolutely in ERM itself. This indicates that *risk rewrite* (i.e., the technique of making the risk estimable from observable data via an equivalent expression) enables ERM in positive-unlabeled learning and is the key of success.

One step further from positive-unlabeled learning is *learning from only U data* without any L data. This is significantly harder than previous learning problems (cf. Figure 1). However, we would still like to train *arbitrary binary classifier*, in particular, deep networks (Goodfellow et al., 2016). Note that for this purpose clustering is suboptimal for two major reasons. First, successful translation of clusters into meaningful classes completely relies on the critical assumption that *one cluster exactly*

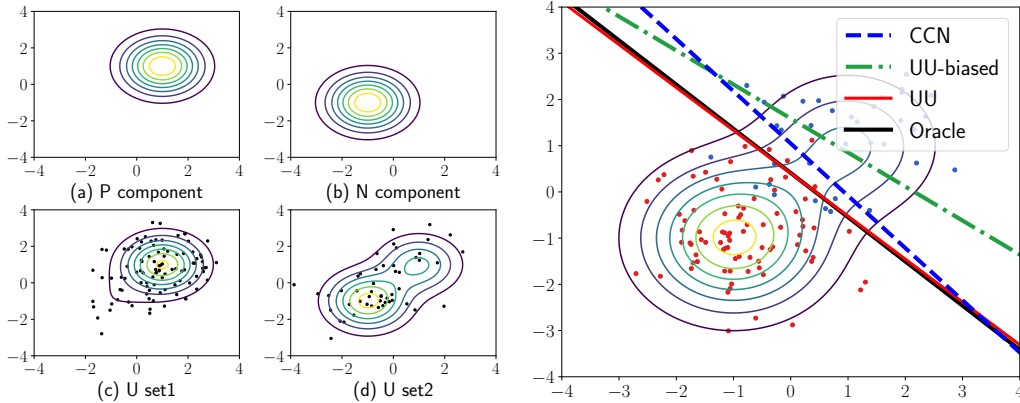

In the left panel, (a) and (b) show positive (P) and negative (N) components of the Gaussian mixture; (c) and (d) show two distributions (with class priors 0.9 and 0.4) where U training data are drawn (marked as black points). The right panel shows the test distribution (with class prior 0.3) and data (marked as blue for P and red for N), as well as four learned classifiers. In the legend, "CCN" refers to Natarajan et al. (2013), "UU-biased" means supervised learning taking larger-/smaller-class-prior U data as P/N data, "UU" is the proposed method, and "Oracle" means supervised learning from the same amount of L data. See Appendix B for more information. We can see that UU is almost identical to Oracle and much better than the other two methods.

Figure 1: Illustrative example of classification from a Gaussian mixture dataset.

*corresponds to one class*, and hence even perfect clustering might still result in poor classification. Second, clustering must introduce additional geometric or information-theoretic assumptions upon which the learning objectives of clustering are built (e.g., Xu et al., 2004; Gomes et al., 2010). As a consequence, we prefer ERM to clustering and then no more assumption is required.

The difficulty is how to estimate the risk from only U data, and our solution is again ERM-enabling risk rewrite in the aforementioned two-step approach. The first step should lead to an *unbiased risk estimator* that will be used in the second step. Subsequently, we can evaluate the empirical training and/or validation risk by plugging only U training/validation data into the risk estimator. Thus, this two-step ERM needs *no L validation data for hyperparameter tuning*, which is a huge advantage in training deep models nowadays. Note that given only U data, by no means could we learn the class priors (Menon et al., 2015), so that we assume *all necessary class priors are also given*. This is the unique type of supervision we will leverage throughout this paper, and hence this learning problem still belongs to weakly-supervised learning rather than unsupervised learning.

In this paper, we raise a fundamental question in weakly-supervised learning—how many sets of U data with *different class priors* are necessary for rewriting the risk? Our answer has two aspects:

- Risk rewrite is impossible given a single set of U data (see Theorem 2 in Sec. 3);
- Risk rewrite becomes possible given two sets of U data (see Theorem 4 in Sec. 4).

This suggests that *three class priors[1] are all you need* to train deep models from only U data, while any two[2] should not be enough. The impossibility is a proof by contradiction, and the possibility is a proof by construction, following which we explicitly design an unbiased risk estimator. Therefore, with the help of this risk estimator, we propose an ERM-based learning method from two sets of U data. Thanks to the unbiasedness of our risk estimator, we derive an *estimation error bound* which certainly guarantees the *consistency of learning* (Mohri et al., 2012; Shalev-Shwartz & Ben-David, 2014).[3] Experiments demonstrate that the proposed method could train multilayer perceptron, All-ConvNet (Springenberg et al., 2015) and ResNet (He et al., 2016) from two sets of U data; it could outperform state-of-the-art methods for learning from two sets of U data. See Figure 1 for how the proposed method works on a Gaussian mixture of two components.

---

[1]Two class-prior probabilities are of the training distributions and one is of the test distribution.

[2]One of the training distribution and one of the test distribution, or two of the training distributions.

[3]Learning is consistent (more specifically the learned classifier is asymptotically consistent), if and only if as the amount of training data approaches infinity, the risk of the learned classifier converges to the risk of the optimal classifier, where the optimality is defined over a given hypothesis class.

## 2 PROBLEM SETTING AND RELATED WORK

Consider the binary classification problem. Let $X$ and $Y$ be the input and output random variables such that

- $p(x, y)$ is the *underlying joint density*,
- $p_{\mathrm{p}}(x) = p(x \mid Y = +1)$ and $p_{\mathrm{n}}(x) = p(x \mid Y = -1)$ are the *class-conditional densities*,
- $p(x)$ is the *marginal density*, and
- $\pi_{\mathrm{p}} = p(Y = +1)$ is the *class-prior probability*.

**Data generation process** Let $\theta$ and $\theta'$ be two valid class priors such that $\theta \neq \theta'$ (here it does not matter if either $\theta$ or $\theta'$ equals $\pi_{\mathrm{p}}$ or neither of them equals $\pi_{\mathrm{p}}$), and let

$$p_{\mathrm{tr}}(x) = \theta p_{\mathrm{p}}(x) + (1 - \theta)p_{\mathrm{n}}(x), \quad p'_{\mathrm{tr}}(x) = \theta' p_{\mathrm{p}}(x) + (1 - \theta')p_{\mathrm{n}}(x) \tag{1}$$

be the marginal densities from which U training data are drawn. Eq. (1) implies there are $p_{\mathrm{tr}}(x, y)$ and $p'_{\mathrm{tr}}(x, y)$, whose class-conditional densities are same and equal to those of $p(x, y)$, and whose class priors are different, i.e.,

$$p_{\mathrm{tr}}(x \mid y) = p'_{\mathrm{tr}}(x \mid y) = p(x \mid y), \quad p_{\mathrm{tr}}(Y = +1) = \theta \neq \theta' = p'_{\mathrm{tr}}(Y = +1).$$

If we could sample L data from $p_{\mathrm{tr}}(x, y)$ or $p'_{\mathrm{tr}}(x, y)$, it would reduce to supervised learning under *class-prior change* (Quiñonero-Candela et al., 2009).

Nonetheless, the problem of interest belongs to weakly-supervised learning—U training (and validation) data are supposed to be drawn according to (1). More specifically, we have

$$\mathcal{X}_{\mathrm{tr}} = \{x_1, \ldots, x_n\} \sim p_{\mathrm{tr}}(x), \quad \mathcal{X}'_{\mathrm{tr}} = \{x'_1, \ldots, x'_{n'}\} \sim p'_{\mathrm{tr}}(x), \tag{2}$$

where $n$ and $n'$ are two natural numbers as the sample sizes of $\mathcal{X}_{\mathrm{tr}}$ and $\mathcal{X}'_{\mathrm{tr}}$. This is exactly same as du Plessis et al. (2013) and Menon et al. (2015) with some different names. In Menon et al. (2015), $\theta$ and $\theta'$ are called *corruption parameters*, and if we assume $\theta > \theta'$, $p_{\mathrm{tr}}(x)$ is called the *corrupted P density* and $p'_{\mathrm{tr}}(x)$ is called the *corrupted N density*. Despite the same data generation process in (2), a vital difference between the problem settings is performance measures to be optimized.

**Performance measures** Let $g : \mathbb{R}^d \to \mathbb{R}$ be an arbitrary *decision function*, i.e., $g$ may literally be any binary classifier. Let $\ell : \mathbb{R} \to \mathbb{R}$ be the *loss function*, such that the value $\ell(z)$ means the loss by predicting $g(x)$ when the ground truth is $y$ where $z = yg(x)$ is the *margin*. The *risk* of $g$ is

$$R(g) = \mathbb{E}_{(X,Y) \sim p(x,y)}[\ell(Yg(X))] = \pi_{\mathrm{p}}\mathbb{E}_{\mathrm{p}}[\ell(g(X))] + (1 - \pi_{\mathrm{p}})\mathbb{E}_{\mathrm{n}}[\ell(-g(X))], \tag{3}$$

where $\mathbb{E}_{\mathrm{p}}[\cdot]$ means $\mathbb{E}_{X \sim p_{\mathrm{p}}}[\cdot]$ and $\mathbb{E}_{\mathrm{n}}[\cdot]$ means $\mathbb{E}_{X \sim p_{\mathrm{n}}}[\cdot]$ respectively. If $\ell$ is the *zero-one loss* that is defined by $\ell_{01}(z) = (1 - \mathrm{sign}(z))/2$, the risk is also known as the *classification error* and it is the standard performance measure in classification. A balanced version of Eq. (3) is

$$B(g) = \frac{1}{2}\mathbb{E}_{\mathrm{p}}[\ell(g(X))] + \frac{1}{2}\mathbb{E}_{\mathrm{n}}[\ell(-g(X))], \tag{4}$$

and if $\ell$ is $\ell_{01}$, (4) is named the *balanced error* (Brodersen et al., 2010). The vital difference is that (3) is chosen in the current paper whereas (4) is chosen in du Plessis et al. (2013) and Menon et al. (2015) as the performance measure to be optimized.

We argue that (3) is more natural as the performance measure for binary classification than (4). By the phrase "binary classification", we mean $\pi_{\mathrm{p}}$ is neither very large nor very small. Otherwise, due to extreme values of $\pi_{\mathrm{p}}$ (i.e., either $\pi_{\mathrm{p}} \approx 0$ or $\pi_{\mathrm{p}} \approx 1$), the problem under consideration should be retrieval or detection rather than binary classification. Hence, it may be misleading to optimize (4), unless $\pi_{\mathrm{p}} \approx \frac{1}{2}$ which implies that Eqs. (3) and (4) are essentially equivalent.

**Related work** Learning from only U data is previously regarded as *discriminative clustering* (Xu et al., 2004; Valizadegan & Jin, 2006; Li et al., 2009; Gomes et al., 2010; Sugiyama et al., 2014; Hu et al., 2017). Their goals are to maximize the margin or the mutual information between $X$ and $Y$. Recall that clustering is suboptimal, since it requires the *cluster assumption* (Chapelle et al., 2002) and it is rarely satisfied in practice that one cluster exactly corresponds to one class.

As mentioned earlier, learning from two sets of U data is already studied in du Plessis et al. (2013) and Menon et al. (2015). Both of them adopt (4) as the performance measure. In the former paper, $g$ is learned by estimating $\mathrm{sign}(p_{\mathrm{tr}}(x) - p'_{\mathrm{tr}}(x))$. In the latter paper, $g$ is learned by taking noisy L data from $p_{\mathrm{tr}}(x)$ and $p'_{\mathrm{tr}}(x)$ as clean L data from $p_{\mathrm{p}}(x)$ and $p_{\mathrm{n}}(x)$, and then its threshold is moved to the correct value by post-processing. In summary, instead of ERM, they evidence the possibility of *empirical balanced risk minimization*, and no impossibility is proven.

Our findings are compatible with *learning from label proportions* (Quadrianto et al., 2009; Yu et al., 2013). Quadrianto et al. (2009) proves that the minimal number of U sets is equal to the number of classes. However, their finding only holds for the linear model, the logistic loss, and their proposed method based on mean operators. On the other hand, Yu et al. (2013) is not ERM-based; it is based on discriminative clustering together with expectation regularization (Mann & McCallum, 2007).

At first glance, our data generation process, using the names from Menon et al. (2015), looks quite similar to *class-conditional noise* (CCN, Angluin & Laird, 1988) in *learning with noisy labels* (cf. Natarajan et al., 2013).[4] In fact, Menon et al. (2015) makes use of *mutually contaminated distributions* (MCD, Scott et al., 2013) that is more general than CCN. Denote by $\tilde{y}$ and $\tilde{p}(\cdot)$ the corrupted label and distributions. Then, CCN and MCD are defined by

$$\begin{pmatrix} \tilde{p}(\tilde{Y} = +1 \mid x) \\ \tilde{p}(\tilde{Y} = -1 \mid x) \end{pmatrix} = T_{\mathrm{CCN}} \begin{pmatrix} p(Y = +1 \mid x) \\ p(Y = -1 \mid x) \end{pmatrix} \quad \text{and} \quad \begin{pmatrix} \tilde{p}(x \mid \tilde{Y} = +1) \\ \tilde{p}(x \mid \tilde{Y} = -1) \end{pmatrix} = T_{\mathrm{MCD}} \begin{pmatrix} p_{\mathrm{p}}(x) \\ p_{\mathrm{n}}(x) \end{pmatrix},$$

where both of $T_{\mathrm{CCN}}$ and $T_{\mathrm{MCD}}$ are 2-by-2 matrices but $T_{\mathrm{CCN}}$ is column normalized and $T_{\mathrm{MCD}}$ is row normalized. It has been proven in Menon et al. (2015) that CCN is a strict special case of MCD. To be clear, $\tilde{p}(\tilde{y})$ is fixed in CCN once $\tilde{p}(\tilde{y} \mid x)$ is specified while $\tilde{p}(\tilde{y})$ is free in MCD after $\tilde{p}(x \mid \tilde{y})$ is specified. Furthermore, $\tilde{p}(x) = p(x)$ in CCN but $\tilde{p}(x) \neq p(x)$ in MCD. Due to this *covariate shift*, CCN methods do not fit MCD problem setting, though MCD methods fit CCN problem setting. To the best of our knowledge, the proposed method is the first MCD method based on ERM.

## 3 LEARNING FROM ONE SET OF U DATA

From now on, we prove that knowing $\pi_{\mathrm{p}}$ and $\theta$ is insufficient for rewriting $R(g)$.

### 3.1 A BRIEF REVIEW OF ERM

To begin with, we review ERM (Vapnik, 1998) by imaging that we are given $\mathcal{X}_{\mathrm{p}} = \{x_1, \ldots, x_n\} \sim p_{\mathrm{p}}(x)$ and $\mathcal{X}_{\mathrm{n}} = \{x'_1, \ldots, x'_{n'}\} \sim p_{\mathrm{n}}(x)$. Then, we would go through the following procedure:

1. Choose a *surrogate loss* $\ell(z)$, so that $R(g)$ in Eq. (3) is defined.
2. Choose a model $\mathcal{G}$, so that $\min_{g \in \mathcal{G}} R(g)$ is achievable by ERM.
3. Approximate $R(g)$ by

$$\widehat{R}_{\mathrm{pn}}(g) = \frac{\pi_{\mathrm{p}}}{n} \sum_{i=1}^{n} \ell(g(x_i)) + \frac{1 - \pi_{\mathrm{p}}}{n'} \sum_{j=1}^{n'} \ell(-g(x'_j)). \tag{5}$$

4. Minimize $\widehat{R}_{\mathrm{pn}}(g)$, with appropriate regularization, by favorite optimization algorithm.

Here, $\ell$ should be *classification-calibrated* (Bartlett et al., 2006),[5] in order to guarantee that $R(g; \ell)$ and $R(g; \ell_{01})$ have the same minimizer over all measurable functions. This minimizer is the *Bayes optimal classifier* and denoted by $g^{**} = \arg\min_g R(g)$. The Bayes optimal risk $R(g^{**})$ is usually unachievable by ERM as $n, n' \to \infty$. That is why by choosing a model $\mathcal{G}$, $g^* = \arg\min_{g \in \mathcal{G}} R(g)$ became the target (i.e., $\widehat{g}_{\mathrm{pn}} = \arg\min_{g \in \mathcal{G}} \widehat{R}_{\mathrm{pn}}(g)$ will converge to $g^*$ as $n, n' \to \infty$). In statistical learning, the *approximation error* is $R(g^*) - R(g^{**})$, and the *estimation error* is $R(\widehat{g}_{\mathrm{pn}}) - R(g^*)$. Learning is consistent if and only if the estimation error converges to zero as $n, n' \to \infty$.

---

[4]There are quite few instance-dependent noise models (Menon et al., 2016; Cheng et al., 2017), and others explore instance-independent noise models (Natarajan et al., 2013; Sukhbaatar et al., 2015; Menon et al., 2015; Liu & Tao, 2016; Goldberger & Ben-Reuven, 2017; Patrini et al., 2017; Han et al., 2018a) or assume no noise model at all (Reed et al., 2015; Jiang et al., 2018; Ren et al., 2018; Han et al., 2018b).

[5]$\ell$ is classification-calibrated if and only if there is a convex, invertible, and nondecreasing transformation $\psi_\ell$ with $\psi_\ell(0) = 0$, such that $\psi_\ell(R(g; \ell_{01}) - \inf_g R(g; \ell_{01})) \leq R(g; \ell) - \inf_g R(g; \ell)$. If $\ell$ is a convex loss, it is classification-calibrated if and only if it is differentiable at the origin and $\ell'(0) < 0$.

## 3.2 IMPOSSIBILITY OF RISK REWRITE

Recall that $R(g)$ is approximated by (5) given $\mathcal{X}_{\mathrm{p}}$ and $\mathcal{X}_{\mathrm{n}}$, which does not work given $\mathcal{X}_{\mathrm{tr}}$ and $\mathcal{X}'_{\mathrm{tr}}$. We might rewrite $R(g)$ so that it could be approximated given $\mathcal{X}_{\mathrm{tr}}$ and/or $\mathcal{X}'_{\mathrm{tr}}$. This is known as the *backward correction* in learning with noisy/corrupted labels (Patrini et al., 2017; see also Natarajan et al., 2013; van Rooyen & Williamson, 2018).

**Definition 1.** *We say that $R(g)$ in (3) is rewritable given $p_{\mathrm{tr}}$, if and only if*[6] *there exist constants $a$ and $b$, such that for any $g$ it holds that*

$$R(g) = \mathbb{E}_{p_{\mathrm{tr}}}[\bar{\ell}(g(X))], \tag{6}$$

*where $\mathbb{E}_{p_{\mathrm{tr}}}[\cdot]$ means $\mathbb{E}_{X \sim p_{\mathrm{tr}}}[\cdot]$ and $\bar{\ell}(z) = a\ell(z) + b\ell(-z)$ is the corrected loss function.*

In Eq. (6), the expectation is with respect to $p_{\mathrm{tr}}$ and $\theta$ is a free variable in it. The impossibility will be stronger, if $\theta$ is unspecified and allowed to be adjusted according to $\pi_{\mathrm{p}}$.

**Theorem 2.** *Let $\ell$ be $\ell_{01}$, or any bounded surrogate loss satisfying that*

$$0 \le \ell(+\infty) = \lim_{z \to +\infty} \ell(z) < \lim_{z \to -\infty} \ell(z) = \ell(-\infty) < +\infty. \tag{7}$$

*Assume $p_{\mathrm{p}}$ and $p_{\mathrm{n}}$ are almost surely separable. Then, $R(g)$ is not rewritable though $\theta$ is free.*[7]

This theorem shows that under the separability assumption of $p_{\mathrm{p}}$ and $p_{\mathrm{n}}$, $R(g)$ is not rewritable. As a consequence, we lack a learning objective, that is, the empirical training risk. It is even worse—we cannot access the empirical validation risk of $g$ after it is trained by other learning methods such as discriminative clustering. In particular, $\ell_{01}$ satisfies (7), which implies that the common practice of hyperparameter tuning is disabled by Theorem 2, since U validation data are also drawn from $p_{\mathrm{tr}}$.

## 4 LEARNING FROM TWO SETS OF U DATA

From now on, we prove that knowing $\pi_{\mathrm{p}}$, $\theta$ and $\theta'$ is sufficient for rewriting $R(g)$.

### 4.1 POSSIBILITY OF RISK REWRITE, AND UNBIASED RISK ESTIMATORS

We have proven that $R(g)$ is not rewritable given $p_{\mathrm{tr}}$, and Quadrianto et al. (2009) has proven that $R(g)$ can be estimated from $\mathcal{X}_{\mathrm{tr}}$ and $\mathcal{X}'_{\mathrm{tr}}$, where $g$ is a linear model and $\ell$ is the logistic loss. These facts motivate us to investigate the possibility of rewriting $R(g)$, where $g$ and $\ell$ are both arbitrary.[8]

**Definition 3.** *We say that $R(g)$ is rewritable given $p_{\mathrm{tr}}$ and $p'_{\mathrm{tr}}$, if and only if*[9] *there exist constants $a$, $b$, $c$ and $d$, such that for any $g$ it holds that*

$$R(g) = \mathbb{E}_{p_{\mathrm{tr}}}[\bar{\ell}_+(g(X))] + \mathbb{E}_{p'_{\mathrm{tr}}}[\bar{\ell}_-(-g(X))], \tag{8}$$

*where $\bar{\ell}_+(z) = a\ell(z) + b\ell(-z)$ and $\bar{\ell}_-(z) = c\ell(z) + d\ell(-z)$ are the corrected loss functions.*

In Eq. (8), the expectations are with respect to $p_{\mathrm{tr}}$ and $p'_{\mathrm{tr}}$ that are regarded as the corrupted $p_{\mathrm{p}}$ and $p_{\mathrm{n}}$. There are two free variables $\theta$ and $\theta'$ in $p_{\mathrm{tr}}$ and $p'_{\mathrm{tr}}$. The possibility will be stronger, if $\theta$ and $\theta'$ are already specified and disallowed to be adjusted according to $\pi_{\mathrm{p}}$.

**Theorem 4.** *Fix $\theta$ and $\theta'$. Assume $\theta > \theta'$; otherwise, swap $p_{\mathrm{tr}}$ and $p'_{\mathrm{tr}}$ to make sure $\theta > \theta'$. Then, $R(g)$ is rewritable, by letting*

$$a = \frac{(1 - \theta')\pi_{\mathrm{p}}}{\theta - \theta'}, \quad b = -\frac{\theta'(1 - \pi_{\mathrm{p}})}{\theta - \theta'}, \quad c = \frac{\theta(1 - \pi_{\mathrm{p}})}{\theta - \theta'}, \quad d = -\frac{(1 - \theta)\pi_{\mathrm{p}}}{\theta - \theta'}. \tag{9}$$

---

[6]This is because the backward correction in (6), if exists, would be unique.

[7]Please find in Appendix A the proofs of theorems.

[8]The technique that underlies Theorem 4 is totally different from Quadrianto et al. (2009). We shall obtain (9) by solving a linear system resulted from Definition 3. As previously mentioned, Quadrianto et al. (2009) is based on mean operators, and it cannot be further generalized to handle nonlinear $g$ or arbitrary $\ell$.

[9]This is similar because the backward correction in (8), if exists, would be unique.

Theorem (4) immediately leads to an unbiased risk estimator, namely,

$$
\begin{aligned}
\widehat{R}_{\mathrm{uu}}(g) = &\frac{1}{n} \sum_{i=1}^{n} \left( \frac{(1-\theta')\pi_{\mathrm{p}}}{\theta - \theta'} \ell(g(x_i)) - \frac{\theta'(1-\pi_{\mathrm{p}})}{\theta - \theta'} \ell(-g(x_i)) \right) \\
&+ \frac{1}{n'} \sum_{j=1}^{n'} \left( -\frac{(1-\theta)\pi_{\mathrm{p}}}{\theta - \theta'} \ell(g(x'_j)) + \frac{\theta(1-\pi_{\mathrm{p}})}{\theta - \theta'} \ell(-g(x'_j)) \right).
\end{aligned}
\tag{10}
$$

Eq. (10) is useful for both training (by plugging U training data into it) and hyperparameter tuning (by plugging U validation data into it). We hereafter refer to the process of obtaining the empirical risk minimizer of (10), i.e., $\widehat{g}_{\mathrm{uu}} = \arg\min_{g\in\mathcal{G}} \widehat{R}_{\mathrm{uu}}(g)$, as *unlabeled-unlabeled* (UU) *learning*. The proposed UU learning is by nature ERM-based, and consequently $\widehat{g}_{\mathrm{uu}}$ can be obtained by powerful *stochastic optimization* algorithms (e.g., Duchi et al., 2011; Kingma & Ba, 2015).

**Simplification**  Note that (10) may require some efforts to implement. Fortunately, it can be simplified by employing $\ell$ that satisfies a *symmetric condition*:

$$
\ell(z) + \ell(-z) = 1. \tag{11}
$$

Eq. (11) covers $\ell_{01}$, a ramp loss $\ell_{\mathrm{ramp}}(z) = \max\{0, \min\{1, (1-z)/2\}\}$ in du Plessis et al. (2014) and a sigmoid loss $\ell_{\mathrm{sig}}(z) = 1/(1 + \exp(z))$ in Kiryo et al. (2017). With the help of (11), (10) can be simplified as

$$
\widehat{R}_{\mathrm{uu}}^{\mathrm{Sym}}(g) = \frac{1}{n} \sum_{i=1}^{n} \alpha \ell(g(x_i)) + \frac{1}{n'} \sum_{j=1}^{n'} \alpha' \ell(-g(x'_j)) - \frac{\theta'(1-\pi_{\mathrm{p}}) + (1-\theta)\pi_{\mathrm{p}}}{\theta - \theta'}, \tag{12}
$$

where $\alpha = (\theta' + \pi_{\mathrm{p}} - 2\theta'\pi_{\mathrm{p}})/(\theta - \theta')$ and $\alpha' = (\theta + \pi_{\mathrm{p}} - 2\theta\pi_{\mathrm{p}})/(\theta - \theta')$. Similarly, (12) is an unbiased risk estimator, and it is easy to implement with existing codes of cost-sensitive learning.

**Special cases**  Consider some special cases of (10) by specifying $\theta$ and $\theta'$. It is obvious that (10) reduces to (5) for supervised learning, if $\theta = 1$ and $\theta' = 0$. Next, (10) reduces to

$$
\widehat{R}_{\mathrm{pu}}(g) = \frac{1}{n} \sum_{i=1}^{n} \pi_{\mathrm{p}} \ell(g(x_i)) - \frac{1}{n} \sum_{i=1}^{n} \pi_{\mathrm{p}} \ell(-g(x_i)) + \frac{1}{n'} \sum_{j=1}^{n'} \ell(-g(x'_j)),
$$

if $\theta = 1$ and $\theta' = \pi_{\mathrm{p}}$, and we recover the unbiased risk estimator in positive-unlabeled learning (du Plessis et al., 2015; Kiryo et al., 2017). Additionally, (10) reduces to a fairly complicated unbiased risk estimator in similar-unlabeled learning (Bao et al., 2018), if $\theta = \pi_{\mathrm{p}}, \theta' = \pi_{\mathrm{p}}^2/(2\pi_{\mathrm{p}}^2 - 2\pi_{\mathrm{p}} + 1)$ or vice versa. Therefore, UU learning is a very general framework in weakly-supervised learning.

## 4.2  Consistency and convergence rate

The consistency of UU learning is guaranteed due to the unbiasedness of (10). In what follows, we analyze the estimation error $R(\widehat{g}_{\mathrm{uu}}) - R(g^*)$ (see Sec. 3.1 for the definition). To this end, assume there are $C_g > 0$ and $C_\ell > 0$ such that $\sup_{g\in\mathcal{G}} \|g\|_\infty \le C_g$ and $\sup_{|z|\le C_g} \ell(z) \le C_\ell$, and assume $\ell(z)$ is Lipschitz continuous for all $|z| \le C_g$ with a Lipschitz constant $L_\ell$. Let $\mathfrak{R}_n(\mathcal{G})$ and $\mathfrak{R}'_{n'}(\mathcal{G})$ be the *Rademacher complexity* of $\mathcal{G}$ over $p_{\mathrm{tr}}(x)$ and $p'_{\mathrm{tr}}(x)$ (Mohri et al., 2012; Shalev-Shwartz & Ben-David, 2014). For convenience, denote by $\chi_{n,n'} = \alpha/\sqrt{n} + \alpha'/\sqrt{n'}$.

**Theorem 5.** *For any $\delta > 0$, let $C_\delta = \sqrt{(\ln 2/\delta)/2}$, then we have with probability at least $1 - \delta$,*

$$
R(\widehat{g}_{\mathrm{uu}}) - R(g^*) \le 4L_\ell \alpha \mathfrak{R}_n(\mathcal{G}) + 4L_\ell \alpha' \mathfrak{R}'_{n'}(\mathcal{G}) + 2C_\ell C_\delta \chi_{n,n'}, \tag{13}
$$

*where the probability is over repeated sampling of $\mathcal{X}_{\mathrm{tr}}$ and $\mathcal{X}'_{\mathrm{tr}}$ for training $\widehat{g}_{\mathrm{uu}}$.*

Theorem 5 ensures that UU learning is consistent (and so are all the special cases): as $n, n' \to \infty$, $R(\widehat{g}_{\mathrm{uu}}) \to R(g^*)$, since $\mathfrak{R}_n(\mathcal{G}), \mathfrak{R}'_{n'}(\mathcal{G}) \to 0$ for all parametric models with a bounded norm such as deep networks trained with weight decay. Moreover, $R(\widehat{g}_{\mathrm{uu}}) \to R(g^*)$ in $\mathcal{O}_p(\chi_{n,n'})$, where $\mathcal{O}_p$ denotes the order in probability, for all linear-in-parameter models with a bounded norm, including non-parametric kernel models in *reproducing kernel Hilbert spaces* (Schölkopf & Smola, 2001).

## 5 EXPERIMENTS

In this section, we experimentally analyze the proposed method in training deep networks and subsequently experimentally compare it with state-of-the-art methods for learning from two sets of U data. The implementation in our experiments is based on Keras (see https://keras.io); it is available at https://github.com/lunanbit/UUlearning.

### 5.1 TRAINING DEEP NEURAL NETWORKS ON BENCHMARKS

In order to analyze the proposed method, we compare it with three supervised baseline methods:

- *small PN* means supervised learning from 10% L data;
- *PN oracle* means supervised learning from 100% L data;
- *small PN prior-shift* means supervised learning from 10% L data under class-prior change.

Notice that the first two baselines have L data identically distributed as the test data, which is very advantageous and thus the experiments in this subsection are merely for a proof of concept.

Table 1 summarizes the benchmarks. They are converted into binary classification datasets; please see Appendix C.1 for details. $\mathcal{X}_{\mathrm{tr}}$ and $\mathcal{X}'_{\mathrm{tr}}$ of the same sample size are drawn according to Eq. (1), where $\theta$ and $\theta'$ are chosen as 0.9, 0.1 or 0.8, 0.2. The test data are just drawn from $p(x, y)$.

Table 1 also describes the models and optimizers. In this table, FC refers to *fully connected neural networks*, AllConvNet refers to *all convolutional net* (Springenberg et al., 2015) and ResNet refers to *residual networks* (He et al., 2016); then, SGD is short for *stochastic gradient descent* (Robbins & Monro, 1951) and Adam is short for *adaptive moment estimation* (Kingma & Ba, 2015).

Recall from Sec. 3.1 that after the model and optimizer are chosen, it remains to determine the loss $\ell(z)$. We have compared the sigmoid loss $\ell_{\mathrm{sig}}(z)$ and the logistic loss $\ell_{\mathrm{log}}(z) = \ln(1 + \exp(-z))$, and found that the resulted classification errors are similar; please find the details in Appendix C.2. Since $\ell_{\mathrm{sig}}$ satisfies (11) and is compatible with (12), we shall adopt it as the surrogate loss.

The experimental results are reported in Figure 2, where means and standard deviations of classification errors based on 10 random samplings are shown, and the table of final errors can be found in Appendix C.2. When $\theta = 0.9$ and $\theta' = 0.1$ (cf. the left column), UU is comparable to PN oracle in most cases. When $\theta = 0.8$ and $\theta' = 0.2$ (cf. the right column), UU performs slightly worse but it is still better than small PN baselines. This is because the task becomes harder when $\theta$ and $\theta'$ become closer, which will be intensively investigated next.

**On the closeness of $\theta$ and $\theta'$** It is intuitive that if $\theta$ and $\theta'$ move closer, $\mathcal{X}_{\mathrm{tr}}$ and $\mathcal{X}'_{\mathrm{tr}}$ will be more similar and thus less informative. To investigate this, we test UU and CCN (Natarajan et al., 2013) on MNIST by fixing $\theta$ to 0.9 or 0.8 and gradually moving $\theta'$ from 0.1 to 0.5, and the experimental results are reported in Figure 3. We can see that when $\theta'$ moves closer to $\theta$, UU and CCN become worse, while UU is affected slightly and CCN is affected severely. The phenomenon of UU can be explained by Theorem 5, where the upper bound in (13) is linear in $\alpha$ and $\alpha'$ which, as $\theta' \to \theta$, are inversely proportional to $\theta - \theta'$. On the other hand, the phenomenon of CCN is caused by stronger covariate shift when $\theta'$ moves closer to $\theta$ rather than the difficulty of the task. This illustrates CCN methods do not fit our problem setting, so that we called for some new learning method (i.e., UU).

Table 1: Specification of benchmark datasets, models, and optimization algorithms.

| Dataset | # Train | # Test | # Feature | $\pi_{\mathrm{p}}$ | Model $g(x; \theta)$ | Optimizer |
|---|---|---|---|---|---|---|
| MNIST | 60,000 | 10,000 | 784 | 0.49 | FC with ReLU (depth 5) | SGD |
| Fashion-MNIST | 60,000 | 10,000 | 784 | 0.50 | FC with ReLU (depth 5) | SGD |
| SVHN | 100,000 | 26,032 | 3,072 | 0.27 | AllConvNet (depth 12) | Adam |
| CIFAR-10 | 50,000 | 10,000 | 3,072 | 0.60 | ResNet (depth 32) | Adam |

See http://yann.lecun.com/exdb/mnist/ for MNIST (LeCun et al., 1998), https://github.com/zalandoresearch/fashion-mnist for Fashion-MNIST (Xiao et al., 2017), http://ufldl.stanford.edu/housenumbers/ for SVHN (Netzer et al., 2011), as well as https://www.cs.toronto.edu/~kriz/cifar.html for CIFAR-10 (Krizhevsky, 2009).

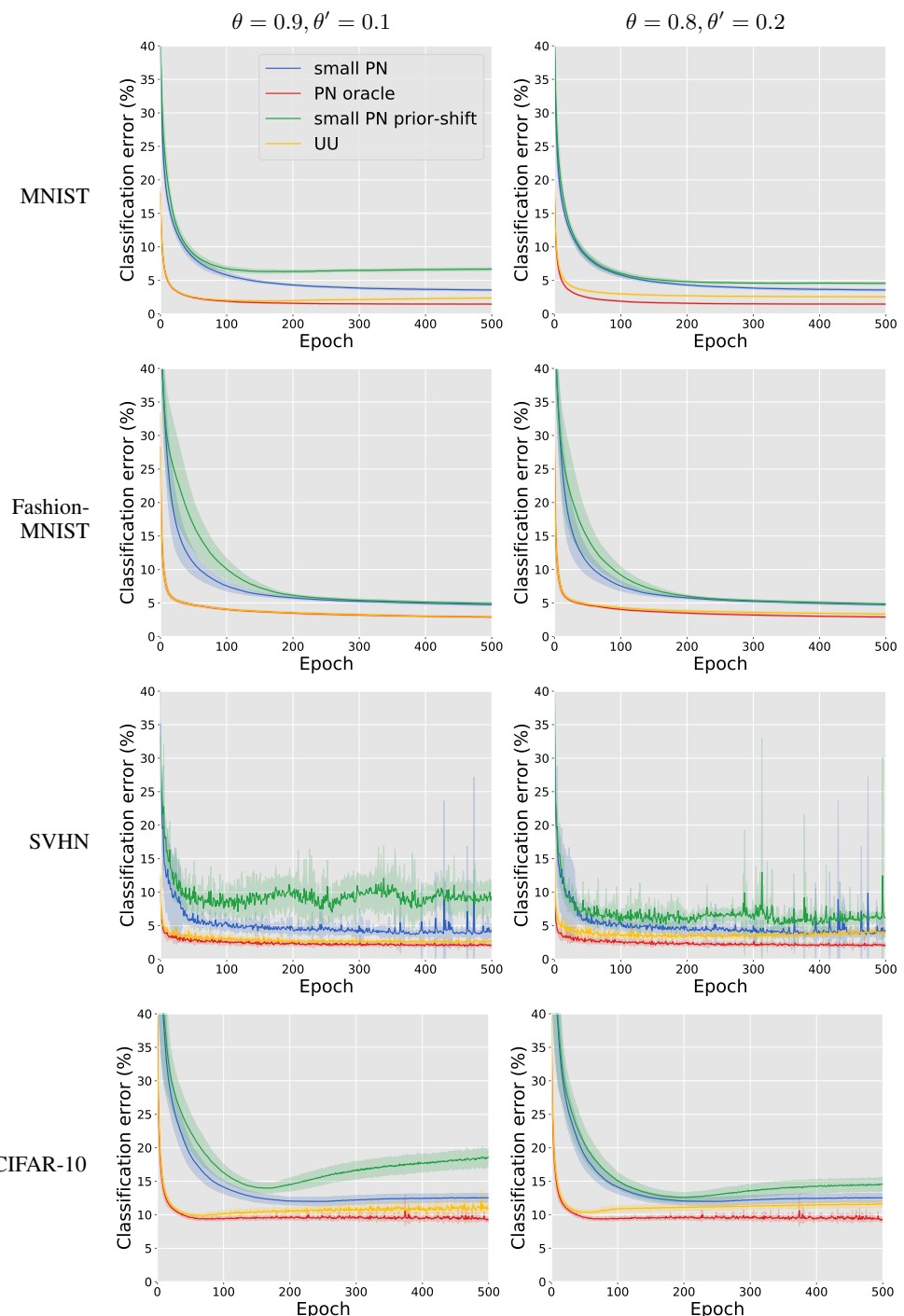

Figure 2: Experimental results of training deep neural networks.

Note that there would be strong covariate shift not only by changing $\theta$ and $\theta'$ but also by changing $n$ and $n'$. The investigation of this issue is deferred to Appendix C.2 due to limited space.

**Robustness against inaccurate training class priors**  Hitherto, we have assumed that the values of $\theta$ and $\theta'$ are accessible, which is rarely satisfied in practice. Fortunately, UU is a robust learning method against inaccurate training class priors. To show this, let $\epsilon$ and $\epsilon'$ be real numbers around 1, $\vartheta = \epsilon\theta$ and $\vartheta' = \epsilon'\theta'$ be perturbed $\theta$ and $\theta'$, and we test UU on MNIST and CIFAR-10 by drawing data using $\theta$ and $\theta'$ but training models using $\vartheta$ and $\vartheta'$ instead. The experimental results in Table 2 imply that UU is fairly robust to inaccurate $\vartheta$ and $\vartheta'$ and can be safely applied in the wild.

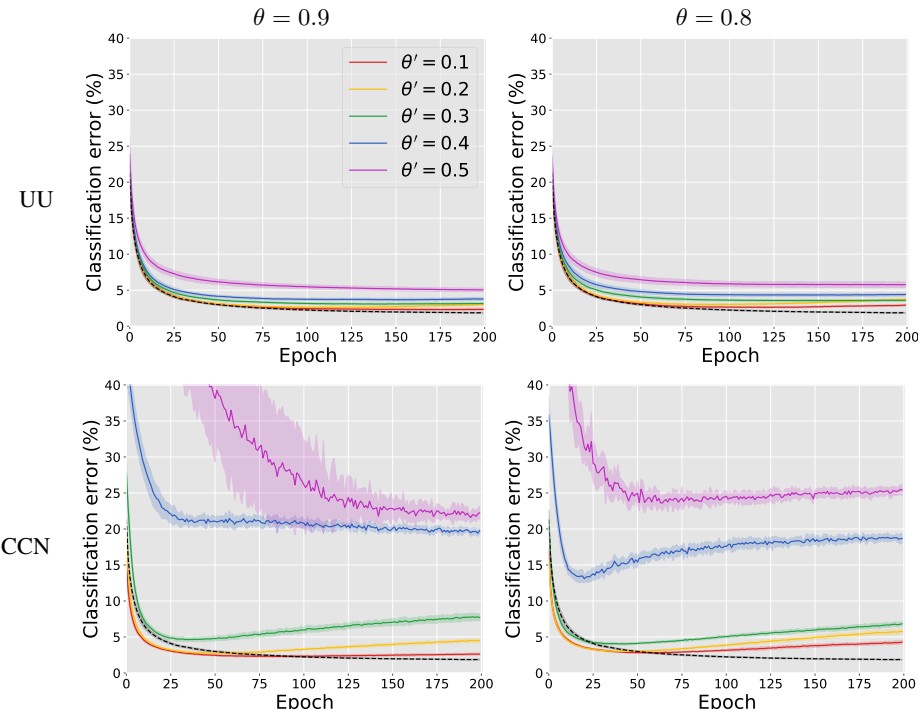

Figure 3: Experimental results of moving $\theta'$ closer to $\theta$ (black dashed lines are PN oracle).

Table 2: Mean errors (standard deviations) in percentage given inaccurate training class priors.

| Dataset | $\theta, \theta'$ | $\epsilon, \epsilon' = 0.8, 0.8$ | $\epsilon, \epsilon' = 0.9, 0.9$ | $\epsilon = \epsilon' = 1.0$ | $\epsilon, \epsilon' = 1.1, 1.1$ | $\epsilon, \epsilon' = 1.2, 1.2$ |
|---------|-------------------|-----------|-----------|-----------|-----------|-----------|
| MNIST | 0.9, 0.1 | 2.31 (0.16) | 2.31 (0.14) | 2.31 (0.14) | 2.32 (0.14) | 2.35 (0.14) |
| | 0.8, 0.2 | 3.00 (0.12) | 3.00 (0.11) | 3.01 (0.10) | 3.02 (0.10) | 3.01 (0.10) |
| | 0.7, 0.3 | 4.24 (0.23) | 4.24 (0.24) | 4.24 (0.26) | 4.25 (0.24) | 4.25 (0.25) |
| CIFAR-10 | 0.9, 0.1 | 10.19 (0.37) | 10.14 (0.29) | 10.14 (0.30) | 10.11 (0.34) | 10.09 (0.35) |
| | 0.8, 0.2 | 10.84 (0.38) | 10.84 (0.40) | 10.77 (0.40) | 10.73 (0.40) | 10.73 (0.40) |
| | 0.7, 0.3 | 12.04 (0.61) | 12.00 (0.54) | 11.92 (0.54) | 11.91 (0.53) | 11.88 (0.53) |

| Dataset | $\theta, \theta'$ | $\epsilon, \epsilon' = 0.8, 1.2$ | $\epsilon, \epsilon' = 0.9, 1.1$ | $\epsilon = \epsilon' = 1.0$ | $\epsilon, \epsilon' = 1.1, 0.9$ | $\epsilon, \epsilon' = 1.2, 0.8$ |
|---------|-------------------|-----------|-----------|-----------|-----------|-----------|
| MNIST | 0.9, 0.1 | 2.30 (0.15) | 2.31 (0.16) | 2.31 (0.14) | 2.30 (0.13) | 2.30 (0.14) |
| | 0.8, 0.2 | 3.00 (0.10) | 3.00 (0.12) | 3.01 (0.10) | 3.02 (0.12) | 3.01 (0.11) |
| | 0.7, 0.3 | 4.19 (0.22) | 4.22 (0.23) | 4.24 (0.26) | 4.24 (0.25) | 4.25 (0.23) |
| CIFAR-10 | 0.9, 0.1 | 10.20 (0.33) | 10.15 (0.34) | 10.14 (0.30) | 10.12 (0.35) | 10.08 (0.37) |
| | 0.8, 0.2 | 10.94 (0.46) | 10.83 (0.39) | 10.77 (0.40) | 10.75 (0.37) | 10.71 (0.43) |
| | 0.7, 0.3 | 12.24 (0.71) | 12.05 (0.59) | 11.92 (0.54) | 11.88 (0.53) | 11.95 (0.49) |

## 5.2 COMPARISON WITH STATE-OF-THE-ART METHODS

Finally, we compare UU with two state-of-the-art methods for dealing with two sets of U data:[10]

- *proportion-SVM* (pSVM, Yu et al., 2013) that is the best in learning from label proportions;
- *balanced error minimization* (BER, Menon et al., 2015) that is the most related work to UU.

The original codes of BER train single-hidden-layer neural networks by LBFGS (which belongs to second-order optimization) in MATLAB. For a fair comparison, we also implement BER by fixing $\pi_\mathrm{p}$ to 0.5 in UU, so that UU and BER only differ in the performance measure. This new baseline is referred to as BER-FC.

---

[10]We downloaded the codes by the original authors; see `https://github.com/felixyu/pSVM` and `https://akmenon.github.io/papers/corrupted-labels/index.html`.

Table 3: Mean errors (standard deviations) in percentage of UU and state-of-the-art methods. Best and comparable methods (paired *t*-test at significance level 1%) are highlighted in boldface.

| Dataset | $\pi_{\mathrm{p}}$ | # Sub | # Train | $\Delta\theta$ | pSVM | BER | BER-FC | UU |
|---|---|---|---|---|---|---|---|---|
| pendigits | 0.1 | 4,000 | 971 | 0.57 | 4.03 (0.27) | 5.51 (1.35) | 5.46 (1.23) | **1.97 (0.78)** |
| covertype | 0.3 | 7,400 | 3,863 | 0.80 | 14.63 (1.00) | 11.33 (0.26) | **5.17 (0.57)** | **4.97 (0.48)** |
| MNIST | 0.5 | 11,800 | 7,139 | 0.77 | N/A | **3.66 (0.20)** | **3.03 (0.25)** | **2.87 (0.28)** |
| spambase | 0.7 | 3,570 | 1,139 | 0.80 | 29.18 (1.29) | **11.28 (1.73)** | 13.98 (1.63) | **12.53 (1.00)** |
| letter | 0.9 | 5,555 | 532 | 0.60 | 15.65 (4.18) | 15.45 (6.99) | 8.45 (2.92) | **3.15 (0.84)** |
| USPS | 0.1 | 4,000 | 971 | 0.57 | 5.91 (1.52) | 12.69 (4.09) | 8.57 (2.40) | **3.74 (1.24)** |
| | 0.3 | 5,000 | 2,605 | 0.80 | 5.55 (0.46) | 5.36 (0.41) | **2.75 (0.28)** | **2.63 (0.18)** |
| | 0.5 | 4,000 | 1,695 | 0.60 | 9.27 (0.61) | **7.27 (1.09)** | **5.48 (1.33)** | 5.52 (1.02) |
| | 0.7 | 5,720 | 1,853 | 0.80 | 8.20 (0.73) | 7.48 (0.65) | **4.23 (0.50)** | **4.43 (0.94)** |
| | 0.9 | 4,445 | 424 | 0.44 | 9.80 (2.07) | 14.13 (2.02) | 18.27 (5.17) | **6.20 (1.33)** |

The first five datasets come with the original codes of BER and USPS is from `https://cs.nyu.edu/~roweis/data.html`. The rows are arranged according to $\pi_{\mathrm{p}}$. In this table, # Sub means the amount of subsampled L training data, # Train means the amount of generated U training data, and $\Delta\theta$ means $\theta - \theta'$. The cell N/A (in MNIST row and pSVM column) is since pSVM is based on maximum margin clustering and is too slow on MNIST. The task would be harder, if $\pi_{\mathrm{p}}$ is closer to 0.5, or # Train or $\Delta\theta$ is smaller.

The information of datasets can be found in Table 3. We work on small datasets following Menon et al. (2015), because pSVM and BER are not reliant on stochastic optimization and cannot handle larger datasets. Furthermore, in order to try different $\pi_{\mathrm{p}}$, we first subsample the original datasets to match the desired $\pi_{\mathrm{p}}$ and then calculate the sample sizes $n$ and $n'$ according to how many P and N data there are in the subsampled datasets, where $\theta$ and $\theta'$ are set as close to 0.9 and 0.1 as possible. For UU and BER-FC, the model is FC with ReLU of depth 5 and the optimizer is SGD. We repeat this sampling-and-training process 10 times for all learning methods on all datasets.

The experimental results are reported in Table 3, and we can see that UU is always the best method (7 out of 10 cases) or comparable to the best method (3 out of 10 cases). Moreover, the closer $\pi_{\mathrm{p}}$ is to 0.5, the better BER and BER-FC are; however, the closer $\pi_{\mathrm{p}}$ is to 0 or 1, the worse they are, and sometimes they are much worse than pSVM. This is because their goal is to minimize the balanced error instead of the classification error. In our experiments, pSVM falls behind, because it is based on discriminative clustering and is also not designed to minimize the classification error.

## 6 CONCLUSIONS

We focused on training arbitrary binary classifier, ranging from linear to deep models, from only U data by ERM. We proved that risk rewrite as the core of ERM is impossible given a single set of U data, but it becomes possible given two sets of U data with different class priors, after we assumed that all necessary class priors are also given. This possibility led to an unbiased risk estimator, and with the help of this risk estimator we proposed UU learning, the first ERM-based learning method from two sets of U data. Experiments demonstrated that UU learning could successfully train fully connected, all convolutional and residual networks, and it compared favorably with state-of-the-art methods for learning from two sets of U data.

ACKNOWLEDGMENTS

NL was supported by the MEXT scholarship No. 171536. MS was supported by JST CREST JP-MJCR1403. We thank all anonymous reviewers for their helpful and constructive comments on the clarity of two earlier versions of this manuscript.

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

## A PROOFS

In this appendix, we prove all theorems.

### A.1 PROOF OF THEOREM 2

We prove the theorem by contradiction, namely, for any such $p(x, y)$ (with almost surely separable $p_\mathrm{p}$ and $p_\mathrm{n}$), for all $a, b$ and all $\theta$, we are able to find some $g$ for which (6) fails. Our argument goes from the special case of $\ell_{01}$ to the general case of $\ell$ satisfying (7).

Firstly, let $g(x) = +\infty$ identically, so that $\ell(g(x)) = 0$ and $\ell(-g(x)) = 1$. Plugging them into (3) and (6), we obtain that

$$b = 1 - \pi_\mathrm{p}.$$

Secondly, let $g(x) = -\infty$ identically; this time $\ell(g(x)) = 1$ and $\ell(-g(x)) = 0$, and we obtain that

$$a = \pi_\mathrm{p}.$$

Thirdly, let $g(x) = +\infty$ over $p_\mathrm{p}$ and $g(x) = -\infty$ over $p_\mathrm{n}$. To be precise, define

$$g(x) = \begin{cases} +\infty, & p_\mathrm{p}(x) > 0 \text{ and } p_\mathrm{n}(x) = 0, \\ -\infty, & p_\mathrm{p}(x) = 0 \text{ and } p_\mathrm{n}(x) > 0, \\ 0, & p_\mathrm{p}(x) > 0 \text{ and } p_\mathrm{n}(x) > 0. \end{cases}$$

This is possible because $g$ is arbitrary. The last case $g(x) = 0$ should have a zero probability, since $p_\mathrm{p}$ and $p_\mathrm{n}$ are almost surely separable. Hence, we have $\ell(g(x)) = 0$ and $\ell(-g(x)) = 1$ over $p_\mathrm{p}$ and $\ell(g(x)) = 1$ and $\ell(-g(x)) = 0$ over $p_\mathrm{n}$, resulting in

$$\begin{aligned} 0 &= \pi_\mathrm{p}\mathbb{E}_\mathrm{p}[\ell(g(X))] + (1 - \pi_\mathrm{p})\mathbb{E}_\mathrm{n}[\ell(-g(X))] \\ &= \theta\mathbb{E}_\mathrm{p}[\bar{\ell}(g(X))] + (1 - \theta)\mathbb{E}_\mathrm{n}[\bar{\ell}(g(X))] \\ &= \theta b + (1 - \theta)a. \end{aligned}$$

By solving this equation, we know that

$$\theta = \frac{a}{a - b} = \frac{\pi_\mathrm{p}}{2\pi_\mathrm{p} - 1}. \tag{14}$$

Nevertheless, $0 \le \theta \le 1$ whereas

- $\pi_\mathrm{p}/(2\pi_\mathrm{p} - 1) < 0$, if $0 < \pi_\mathrm{p} < 1/2$;
- $\pi_\mathrm{p}/(2\pi_\mathrm{p} - 1) > 1$, if $1/2 < \pi_\mathrm{p} < 1$;
- $\pi_\mathrm{p}/(2\pi_\mathrm{p} - 1)$ is undefined, if $\pi_\mathrm{p} = 1/2$.

Therefore, (14) must be a contradiction, unless $\pi_\mathrm{p} = 0$ or $\pi_\mathrm{p} = 1$ which implies that there is just a single class and the problem under consideration is not binary classification.

Finally, given any $\ell$ satisfying (7), it is not difficult to verify that the three $g$ above lead to the same contradiction with exactly the same $a, b$ and $\theta$ by solving a bit more complicated equations. □

### A.2 PROOF OF THEOREM 4

Let $J(g)$ be an alias of $R(g)$ in Definition 3 serving as the learning objective, i.e.,

$$J(g) = \mathbb{E}_{p_\mathrm{tr}}[\bar{\ell}_+(g(X))] + \mathbb{E}_{p'_\mathrm{tr}}[\bar{\ell}_-(-g(X))], \tag{15}$$

then

$$\begin{aligned} J(g) &= \mathbb{E}_{p_\mathrm{tr}}[a\ell(g(X)) + b\ell(-g(X))] + \mathbb{E}_{p'_\mathrm{tr}}[c\ell(-g(X)) + d\ell(g(X))] \\ &= \theta\mathbb{E}_\mathrm{p}[a\ell(g(X)) + b\ell(-g(X))] + (1 - \theta)\mathbb{E}_\mathrm{n}[a\ell(g(X)) + b\ell(-g(X))] \\ &\quad + \theta'\mathbb{E}_\mathrm{p}[c\ell(-g(X)) + d\ell(g(X))] + (1 - \theta')\mathbb{E}_\mathrm{n}[c\ell(-g(X)) + d\ell(g(X))] \\ &= (a\theta + d\theta')\mathbb{E}_\mathrm{p}[\ell(g(X))] + (b\theta + c\theta')\mathbb{E}_\mathrm{p}[\ell(-g(X))] \\ &\quad + [a(1 - \theta) + d(1 - \theta')]\mathbb{E}_\mathrm{n}[\ell(g(X))] + [b(1 - \theta) + c(1 - \theta')]\mathbb{E}_\mathrm{n}[\ell(-g(X))]. \end{aligned}$$

On the other hand,

$$J(g) = \pi_{\mathrm{p}} \mathbb{E}_{\mathrm{p}}[\ell(g(X))] + (1 - \pi_{\mathrm{p}}) \mathbb{E}_{\mathrm{n}}[\ell(-g(X))],$$

since $J(g)$ is an alias of $R(g)$. As a result, in order to minimize $R(g)$ in (3), it suffices to minimize $J(g)$ in (15), if we can make

$$
\begin{aligned}
a\theta + d\theta' &= \pi_{\mathrm{p}}, \\
b\theta + c\theta' &= 0, \\
a(1 - \theta) + d(1 - \theta') &= 0, \\
b(1 - \theta) + c(1 - \theta') &= 1 - \pi_{\mathrm{p}}.
\end{aligned}
$$

Solving these equations gives us Eq. (9), which concludes the proof. $\qquad\square$

## A.3 PROOF OF THEOREM 5

First, we show the uniform deviation bound, which is useful to derive the estimation error bound.

**Lemma 6.** *For any $\delta > 0$, let $C_\delta = \sqrt{(\ln 2/\delta)/2}$, then we have with probability at least $1 - \delta$,*

$$\sup_{g \in \mathcal{G}} |\widehat{R}_{\mathrm{uu}}(g) - R(g)| \le 2L_\ell \alpha \mathfrak{R}_n(\mathcal{G}) + 2L_\ell \alpha' \mathfrak{R}'_{n'}(\mathcal{G}) + C_\ell C_\delta \chi_{n,n'}, \qquad (16)$$

*where the probability is over repeated sampling of data for evaluating $\widehat{R}_{\mathrm{uu}}(g)$.*

*Proof.* Consider the one-side uniform deviation $\sup_{g \in \mathcal{G}} \widehat{R}_{\mathrm{uu}}(g) - R(g)$. Since $0 \le \ell(z) \le C_\ell$, the change of it will be no more than $C_\ell \alpha/n$ if some $x_i$ is replaced, or no more than $C_\ell \alpha'/n'$ if some $x'_j$ is replaced. Subsequently, *McDiarmid's inequality* (McDiarmid, 1989) tells us that

$$\Pr\{\sup_{g \in \mathcal{G}} \widehat{R}_{\mathrm{uu}}(g) - R(g) - \mathbb{E}[\sup_{g \in \mathcal{G}} \widehat{R}_{\mathrm{uu}}(g) - R(g)] \ge \epsilon\} \le \exp\left(-\frac{2\epsilon^2}{C_\ell^2(\alpha^2/n + \alpha'^2/n')}\right),$$

or equivalently, with probability at least $1 - \delta/2$,

$$
\begin{aligned}
\sup_{g \in \mathcal{G}} \widehat{R}_{\mathrm{uu}}(g) - R(g) &\le \mathbb{E}[\sup_{g \in \mathcal{G}} \widehat{R}_{\mathrm{uu}}(g) - R(g)] + C_\ell(\alpha/\sqrt{n} + \alpha'/\sqrt{n'})\sqrt{(\ln 2/\delta)/2} \\
&= \mathbb{E}[\sup_{g \in \mathcal{G}} \widehat{R}_{\mathrm{uu}}(g) - R(g)] + C_\ell C_\delta \chi_{n,n'}.
\end{aligned}
$$

By *symmetrization* (Vapnik, 1998), it is a routine work to show that

$$\mathbb{E}[\sup_{g \in \mathcal{G}} \widehat{R}_{\mathrm{uu}}(g) - R(g)] \le 2\alpha \mathfrak{R}_n(\ell \circ \mathcal{G}) + 2\alpha' \mathfrak{R}'_{n'}(\ell \circ \mathcal{G}),$$

and according to *Talagrand's contraction lemma* (Shalev-Shwartz & Ben-David, 2014),

$$\mathfrak{R}_n(\ell \circ \mathcal{G}) \le L_\ell \mathfrak{R}_n(\mathcal{G}), \quad \mathfrak{R}'_{n'}(\ell \circ \mathcal{G}) \le L_\ell \mathfrak{R}'_{n'}(\mathcal{G}).$$

The one-side uniform deviation $\sup_{g \in \mathcal{G}} R(g) - \widehat{R}_{\mathrm{uu}}(g)$ can be bounded similarly. $\qquad\square$

Based on Lemma 6, the estimation error bound (13) is proven through

$$
\begin{aligned}
R(\widehat{g}_{\mathrm{uu}}) - R(g^*) &= \left(\widehat{R}_{\mathrm{uu}}(\widehat{g}_{\mathrm{uu}}) - \widehat{R}_{\mathrm{uu}}(g^*)\right) + \left(R(\widehat{g}_{\mathrm{uu}}) - \widehat{R}_{\mathrm{uu}}(\widehat{g}_{\mathrm{uu}})\right) + \left(\widehat{R}_{\mathrm{uu}}(g^*) - R(g^*)\right) \\
&\le 0 + 2\sup_{g \in \mathcal{G}} |\widehat{R}_{\mathrm{uu}}(g) - R(g)| \\
&\le 4L_\ell \alpha \mathfrak{R}_n(\mathcal{G}) + 4L_\ell \alpha' \mathfrak{R}'_{n'}(\mathcal{G}) + 2C_\ell C_\delta \chi_{n,n'},
\end{aligned}
$$

where $\widehat{R}_{\mathrm{uu}}(\widehat{g}_{\mathrm{uu}}) \le \widehat{R}_{\mathrm{uu}}(g^*)$ by the definition of $\widehat{g}_{\mathrm{uu}}$. $\qquad\square$

## B  Supplementary information on Figure 1

In the introduction, we illustrated the learning problem and the proposed method using a Gaussian mixture of two components. The details of this illustrative example are presented here.

The P component $p_p(x)$ and N component $p_n(x)$ are both two-dimensional Gaussian distributions. Their means are

$$\boldsymbol{\mu}_+ = [+1, +1]^\top, \quad \boldsymbol{\mu}_- = [-1, -1]^\top,$$

and their covariance is the identity matrix. The two training distributions are created following (1) with class priors $\theta = 0.9$ and $\theta' = 0.4$. Subsequently, the two sets of U training data were sampled from those distributions with sample sizes $n = 2000$ and $n' = 1000$. Moreover, $p_p(x)$ and $p_n(x)$ are combined to form the test distribution $p(x, y)$ with weights 0.3 and 0.7, so $\pi_p = 0.3$.

Note that $p(x)$ changes between training and test distributions (which can be seen from Figure 1 by comparing (c) and (d) in the left panel and the right panel). This is the key difference between UU and CCN (Natarajan et al., 2013).

For training, a linear (-in-input) model $g(x) = \boldsymbol{\omega}^\top x + b$ where $\boldsymbol{\omega} \in \mathbb{R}^2$ and $b \in \mathbb{R}$, and a sigmoid loss $\ell_{\mathrm{sig}}(z) = 1/(1 + \exp(z))$ were used. SGD was employed for optimization, where the learning rate was 0.01 and the batch size was 128. The model just has three parameters, so for the sake of a clear comparison of different risk estimators, we did not add any regularization. For every method, the model was trained 500 epochs. The final models are plotted in Figure 1.

## C  Supplementary information on the experiments

### C.1  Setup

**MNIST**  This is a grayscale image dataset of handwritten digits from 0 to 9 where the size of the images is 28*28. It contains 60,000 training images and 10,000 test images. Since it has 10 classes originally, we used the even digits as the P class and the odd digits as the N class, respectively.

The model was FC with ReLU as the activation function: $d$-300-300-300-300-1. Batch normalization (Ioffe & Szegedy, 2015) was applied before hidden layers. An $\ell_2$-regularization was added, where the regularization parameter was fixed to 1e-4. The model was trained by SGD with an initial learning rate 1e-3 and a batch size 128. In addition, the learning rate was decreased by

$$\frac{1}{1 + \mathrm{decay} \cdot \mathrm{epoch}},$$

where decay was chosen from {0, 1e-6, 1e-5, 5e-5, 1e-4, 5e-4}. This is a learning rate schedule built in Keras.

**Fashion-MNIST**  This is also a grayscale image dataset similarly to MNIST, but here each data is associated with a label from 10 fashion item classes. It was converted into a binary classification dataset as follows:

- the P class is formed by 'T-shirt', 'Pullover', 'Coat', 'Shirt', and 'Bag';
- the N class is formed by 'Trouser', 'Dress', 'Sandal', 'Sneaker', and 'Ankle boot'.

The model and optimizer were same as MNIST, except that the initial learning rate was 1e-4.

**SVHN**  This is a 32*32 color image dataset of street view house numbers from 0 to 9. It consists of 73,257 training data, 26,032 test data, and 531,131 extra training data. We sampled 100,000 data for training from the concatenation of training data and extra training data—the extra training data were used to ensure enough training data so as to perform class-prior changes. For SVHN dataset, '0', '6', '8', '9' made up the P class, and '1', '2', '3', '4', '5', '7' made up the N class.

The model was AllConvNet (Springenberg et al., 2015) as follows.

0th (input) layer: (32*32*3)-
1st to 3rd layers: [C(3*3, 96)]*2-C(3*3, 96, 2)-

Table 4: Means (standard deviations) of the final classification errors in percentage corresponding to Figure 2. Best and comparable methods (excluding PN oracle) based on the paired *t*-test at the significance level 1% are highlighted in boldface.

| Dataset | $\theta, \theta'$ | small PN | small PN prior-shift | UU | PN oracle |
|---------|-------------------|----------|----------------------|-----|-----------|
| MNIST | 0.9, 0.1 | 3.56 (0.13) | 6.69 (0.23) | **2.37 (0.17)** | 1.44 (0.08) |
| | 0.8, 0.2 | 3.56 (0.13) | 4.56 (0.16) | **2.55 (0.11)** | |
| Fashion-MNIST | 0.9, 0.1 | 4.76 (0.17) | 4.93 (0.16) | **2.94 (0.07)** | 2.90 (0.10) |
| | 0.8, 0.2 | 4.76 (0.17) | 4.86 (0.16) | **3.35 (0.13)** | |
| SVHN | 0.9, 0.1 | 4.28 (1.07) | 9.26 (2.41) | **2.69 (0.20)** | 2.08 (0.43) |
| | 0.8, 0.2 | **4.28 (1.07)** | 6.16 (0.66) | 3.99 (0.51) | |
| CIFAR-10 | 0.9, 0.1 | 12.53 (0.69) | 18.58 (1.30) | **10.97 (0.91)** | 9.26 (0.41) |
| | 0.8, 0.2 | 12.53 (0.69) | 14.59 (1.05) | **11.64 (0.54)** | |

    4th to 6th layers:  [C(3*3, 192)]*2-C(3*3, 192, 2)-
     7th to 9th layers:  C(3*3, 192)-C(1*1, 192)-C(1*1, 10)-
 10th to 12th layers:  1000-1000-1

where C(3*3, 96) means 96 channels of 3*3 convolutions followed by ReLU, [ · ]*2 means 2 such layers, C(3*3, 96, 2) means a similar layer but with stride 2, etc. Again, batch normalization and $\ell_2$-regularization with a regularization parameter 1e-5 were applied. The optimizer was Adam with the default momentum parameters ($\beta_1 = 0.9$ and $\beta_2 = 0.999$), an initial learning rate 1e-3, and a batch size 500.

**CIFAR-10**    This dataset consists of 60,000 32*32 color images in 10 classes, and there are 5,000 training images and 1,000 test images per class. For CIFAR-10 dataset,

- the P class is composed of 'bird', 'cat', 'deer', 'dog', 'frog' and 'horse';
- the N class is composed of 'airplane', 'automobile', 'ship' and 'truck'.

The model was ResNet-32 (He et al., 2016) as follows.

      0th (input) layer:  (32*32*3)-
    1st to 11th layers:  C(3*3, 16)-[C(3*3, 16), C(3*3, 16)]*5-
 12th to 21st layers:  [C(3*3, 32), C(3*3, 32)]*5-
22nd to 31st layers:  [C(3*3, 64), C(3*3, 64)]*5-
         32nd layer:  Global Average Pooling-1

where [ ·, · ] means a building block (He et al., 2016). The optimization setup was the same as for SVHN, except that the regularization parameter was set to be 5e-3 and the initial learning rate was set to be 1e-5.

**Remark**    In the experiments on the closeness of $\theta$ and $\theta'$ and on the robustness against inaccurate training class priors, we sampled 40,000 training data from all the training data of MNIST in order to make it feasible to perform class-prior changes.

## C.2   Results

**Final classification errors**    Please find in Table 4.

**Comparison of different losses**    We have compared the sigmoid loss $\ell_{\text{sig}}(z)$ and the logistic loss $\ell_{\text{log}}(z)$ on MNIST. The experimental results are reported in Figure 4. We can see that the resulted classification errors are similar—in fact, $\ell_{\text{sig}}(z)$ is a little better.

**On the variation of $n$ and $n'$**    We have further investigated the issue of covariate shift by varying $n$ and $n'$. Likewise, we test UU and CCN on MNIST by fixing $n'$ to 20,000 and gradually moving $n$ from 20,000 to 4,000, where $\theta'$ is fixed to 0.4 and $\theta$ is chosen from 0.9 or 0.8. The experimental results in Figure 5 indicate that when $n$ moves farther from $n'$, UU and CCN become worse, while

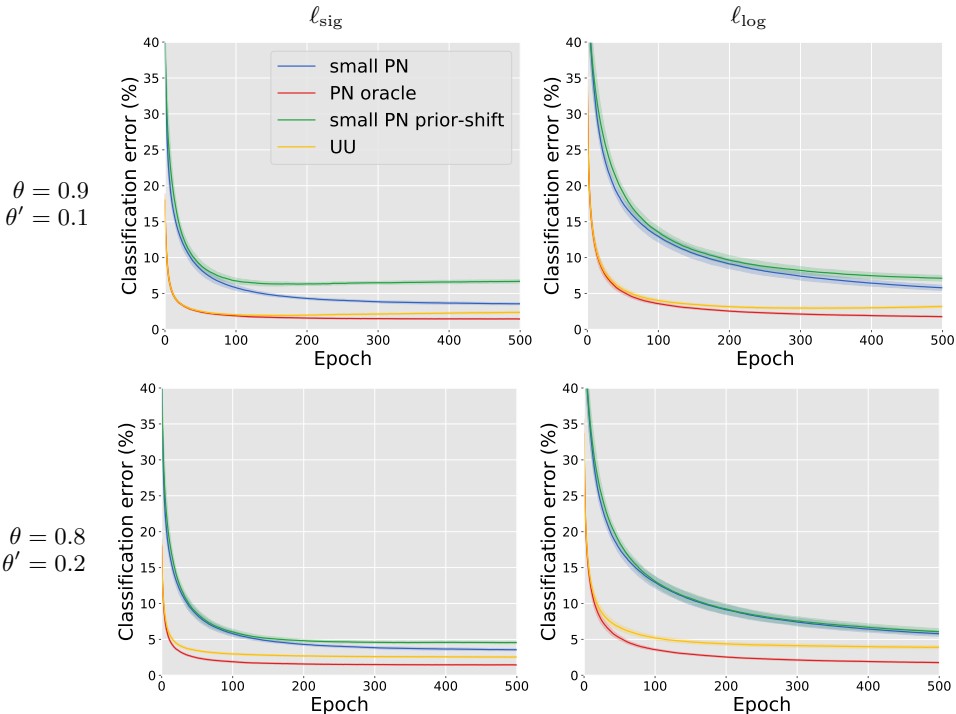

Figure 4: Experimental results of comparing $\ell_{\mathrm{sig}}$ and $\ell_{\mathrm{log}}$.

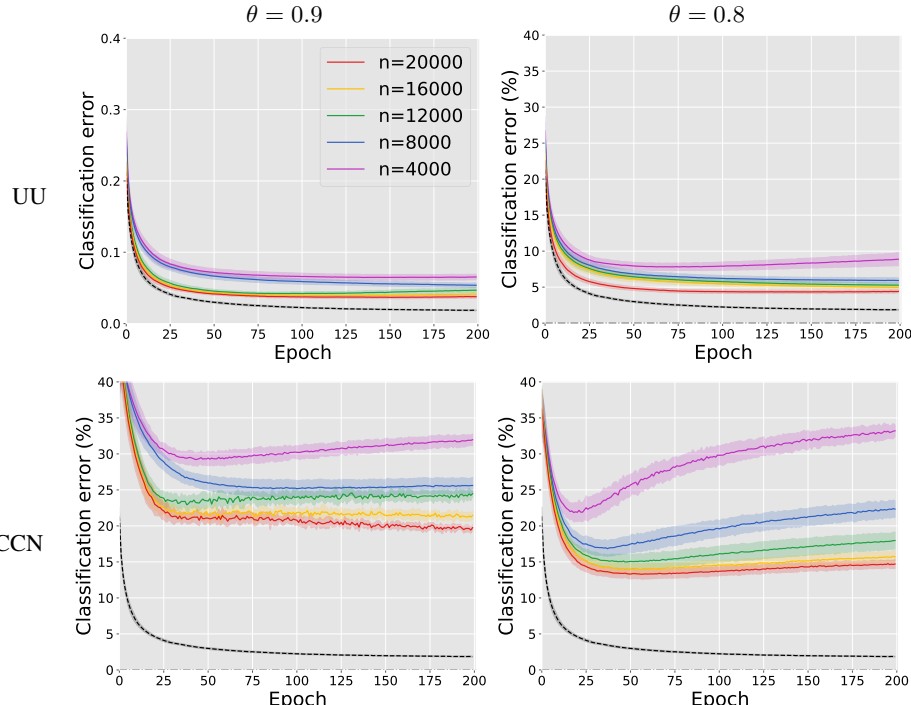

Figure 5: Experimental results of moving $n$ farther from $n'$ (black dashed lines are PN oracle).

UU is affected slightly and CCN is affected severely. Figure 5 is consistent with Figure 3, showing that CCN methods do not fit our problem setting.

