# OpenReview forum: "On the Minimal Supervision for Training Any Binary Classifier from Only Unlabeled Data"
_ICLR.cc/2019/Conference_

### Official Review · AnonReviewer2 · 2018-11-02
**Nice conceptual contribution + thoroughly executed; however dense writing, and in a crowded area**

**Rating:** 7
**Confidence:** 4

**Review:**

This paper studies the weak supervision setting of learning a general binary classifier from two unlabeled (U) datasets with known class balances. The authors establish that this is possible by constructing an unbiased estimator, analyze its convergence theoretically, and then run experiments using modern image classification models.

Pros:
- This work demonstrates, theoretically and empirically, a simple way to train generic models using only the known class balances of several sets of unlabeled data (having the same conditional distributions p(x|y))---a very interesting configuration of weak supervision, an increasingly popular and important area

- The treatment is thorough, proceeding from establishing the minimum number of U datasets, constructing the estimator, analyzing convergence, and implementing thorough experiments

Cons:
- This is a crowded area (as covered in their related work section). As they cite, (Quadrianto et al., 2009) proposed this setting and considered linear models for k-wise classification.  Moreover, the two U datasets with known class balances can equivalently be viewed as two weak / noisy label sources with known accuracies.  Thus this work connects to many areas- both in noisy learning, as they cite heavily, but also in methods (in e.g. crowdsourcing and multi-source weak supervision) where several sources label unlabeled datasets with unknown accuracies (which are often estimated in an unsupervised fashion).

- The overall clarity of the paper's writing could be improved. For example, the introduction and related work sections take up a large portion of the paper, but are very dense and heavy with jargon that is not internally defined upfront; for example "risk rewrite" is introduced in paragraph 2 with no internal definition and then used subsequently throughout the paper (this defn would be simple enough to give: in the context of this paper, "risk rewrite" means a linear combination of the class-conditional losses; or more generally, the expected loss w.r.t. distribution over classes...).  Also intuition could be briefly given about the theorem proof strategies.

- The difference between the two class distributions over the U datasets seems like an important quantity (akin, in e.g. weak supervision / crowd source modeling papers, to quantity of how bounded away from random noise the labelers are). This is treated empirically, but would be stronger to have this show up in the theory somewhere.

- Other prior work here has handled k classes with k U sets; could have extended to cover this setting too, since seems natural

Overall take: This learning from label proportions setting has been covered before, but this paper presents it in an overall clean and general way, testing it empirically on modern models and datasets, which is an interesting contribution.

Other minor points:
- The argument for / distinction between using eqns. (3) and (4) seems a bit ad hoc / informal ("we argue that...").  This is an important point...
- Theorem 1 proof seems fine, but some intuition in the main body would be nice.
- What does "classification calibrated" mean?
- Saying that three U sets are needed, where this includes the test set, seems a bit non-standard?  Also I'm confused- isn't a labeled test set used?  So what is this third U set for?
- The labels l_+ and l_- in Defn. 3 seem to imply that the two U sets are positive vs. negative; but this is not the case, correct…?
- Stating both Lemma 5 and Thm 6 seems unnecessary
- In Fig. 2, seems like could have trained for longer and perhaps some of the losses would have continued decreasing?  In particular, small PN?  Also, a table of the final test set accuracies would have been very helpful.
- More detail on experimental protocol would be helpful: what kind of hyperparameter tuning was done? repeated runs averaging?  It seems odd, for example in Fig. 3, that the green lines are so different in (a) vs. (c), and not in the way that one would expect given the decrease in theta

---

> ### Author Response · Authors · 2018-11-16
> **Thank you so much for your time and constructive comments**
>
> We are still working on the experiments for extending the number of max epochs for training from 200 to 500 and investigating whether standard supervised learning with limited data can be significantly better than now. After that we will go to improve the clarity of the paper following your comments. Please find our detailed responses below.

---

> > ### Author Response · Authors · 2018-11-16
> > **Answers to other minor points**
> >
> > Q: Meaning of “classification calibrated”
> > A: Classification-calibrated loss functions are defined in “P. L. Bartlett, M. I. Jordan, and J. D. McAuliffe. Convexity, classification, and risk bounds. Journal of the American Statistical Association, 2006.” Briefly speaking, using such a surrogate loss guarantees that under mild assumption on the model, the learned model will converge to the Bayes optimal classifier, as the number of training data goes to infinity. Almost all popular losses are classification calibrated---actually, a monotonic and differentiable loss should be classification calibrated, if its gradient at zero is negative.
> >
> > Q: Saying that three U sets are needed, where this includes the test set, seems a bit non-standard
> > A: Sorry for the confusion. To be clear, *three class priors* are needed, i.e. two for the training distributions and one for the test distribution. However, *two U sets* corresponding to two training distributions are needed, and we don’t need any training data from the test distribution.
> >
> > Q: The labels l_+ and l_- in Definition 3 seem to imply the two U sets are positive vs. negative; but this is not the case, correct?
> > A: Yes, the two U sets are completely unlabeled, that is, neither positive nor negative. We followed the notation suggested by the area of learning with noisy labels which caused this confusion. In this context, the U set with larger class prior is regarded as *corrupted positive* data and the U set with smaller class prior is regarded as *corrupted negative* data. We will clearly explain this notation in the revised version.

---

> > ### Author Response · Authors · 2018-11-16
> > **Theoretical analysis of the difference between two U sets for training**
> >
> > This has been explored in Theorem 6, where the estimation error bound of the proposed method is linear in alpha and alpha’. By the definitions of alpha and alpha’, we can see that alpha and alpha’ are both non-negative and no more than 4/(theta-theta’) under the assumption that theta>theta’. Thus, the larger theta-theta’ is, the better the proposed method performs. This theoretical result is consistent with our empirical results shown in Figures 2 and 3.

---

> > ### Author Response · Authors · 2018-11-16
> > **Relationship to learning from label proportions (LLP) and natural extensions to k classes with k U sets**
> >
> > Thanks for the suggestion, but extending this paper to binary or k classes with k U sets is beyond the scope of the current paper. It seems that the problem setting of LLP is more general than this paper since LLP can make use of k U sets to learn a binary classifier. Nevertheless, the goal of learning or the model to be learned should be taken into account too: the goal of LLP is to learn a linear model whereas our goal is to learn either linear or deep model. Existing LLP methods based on ERM, i.e., “N. Quadrianto, A. J. Smola, T. S. Caetano, and Q. V. Le. Estimating labels from label proportions. JMLR, 2009”, cannot learn any nonlinear model.
> >
> > Here we distinguish three cases of extending A to B where A is some component in some existing method. The first case is that A can be naturally extended to B but the authors didn’t know/realize it. The second case is that extending A to B has no problem theoretically but the performance can be quite poor practically. The third case is that A cannot be extended to B theoretically.
> >
> > For example, extending the logistic loss in LLP to any linear-odd loss (i.e., l(z) - l(-z) = -z; see [1]) is the first case. Extending the linear model in many LNL methods to deep models is the second case---the mathematical derivations suggest they are model-independent, but the performances turn out to be quite poor because deep models are much better at memorizing noisy labels than linear models. Finally, extending the linear model in LLP to deep models is the third case---this can be explained by the proof of Theorem 3 in [1], where the key observation is y*g(x)=g(y*x) if g is linear in its parameters; as a result, only the expectation of y*x needs to be estimated that is known as the mean operator and is the technique of LLP. Therefore, LLP cannot benefit from deep learning unless it can get rid of the mean operator.
> >
> > On the other hand, the proposed method shares the technique for designing unbiased risk estimators in LNL (i.e., risk rewrite). For learning a binary classifier, we proved that one U set is not enough but two U sets are enough. However, given (more than) three U sets for training, how to meaningfully incorporate all U sets is an open question. Now we cannot say it belongs to the second or third case, but we are sure it is not the first case.
> >
> > [1] G. Patrini, F. Nielsen, R. Nock, M. Carioni. Loss factorization, weakly supervised learning and label noise robustness. ICML, 2016.

---

> > ### Author Response · Authors · 2018-11-16
> > **Relationship to crowdsourcing and learning with noisy labels (LNL)**
> >
> > Thanks for pointing out that crowdsourcing is also related to this paper! We are conjecturing that data generation processes in crowdsourcing depend heavily on non-expert labelers, while data generation/corruption processes in LNL are more theoretical/statistical. In LNL, when p(y|x) is corrupted, we may first draw x_i from p(x) and then manipulate y_i according to corrupted p(y|x). However, when p(x|y) is corrupted, we have to first pick up y_i and then draw corrupted x_i directly from corrupted p(x|y), since here p(x) is different in clean and corrupted joint densities. In this sense, the problem settings seem not very related. We will carefully check this issue later since we are not very familiar with crowdsourcing. Could you please recommend a few crowdsourcing and multi-source learning papers for our reference?
> >
> > Note that CCN noise model (where p(y|x) is corrupted) has no covariate shift, and MCD noise model (where p(x|y) is corrupted) has inevitable covariate shift. The empirical studies cited in this paper from the computer vision society, if not CCN-based, basically assume that noisy labels are from p(y_noisy|x,y_clean) and thus again has no covariate shift. This paper is the first to experimentally show that unbiased risk estimators originally designed for no-covariate-shift case don’t work in covariate-shift case. As a consequence, this paper is related to but still fairly different from the majority of LNL papers. To the best of our knowledge, this paper is the fourth paper going along this specific direction (after three papers from COLT 2013, TAAI 2013 and ICML 2015).

---

### Official Review · AnonReviewer4 · 2018-11-14
**thorough review of material and refinement that surpasses the state-of-the-art; needs some more development on real-world experiment or justified statement to defer issues until later paper**

**Rating:** 8
**Confidence:** 3

**Review:**

Summary:
The authors introduce the task of learning from unlabeled data clearly and concisely with sufficient reference to background material. They propose a learning approach, called UU, from two unlabeled datasets with known class priors and prove consistency and convergence rates. Their experiments are insightful to the problem, revealing how the two datasets must be sufficiently separated and how UU learning outperforms state-of-the-art approaches. The writing is clear and the idea is an original refinement of earlier work, justified by its exceeding state-of-the-art approaches. However, the paper needs more experimentation.

Further details:
While the introduction and set-up is long, it positions the paper well by making it approachable to someone not directly in the subject area and delineating how the approach differs from existing theory. The paper flows smoothly and the arguments build sequentially. A few issues are left unaddressed:
- How does the natural extension of UU learning extend beyond the binary setting?
- As the authors state, in the wild the class priors may not be known. Their experiment is not completely satisfying because it scales both priors the same. It would be more interesting to experimentally consider them with two different unknown error rates. If this were theoretically addressed (even under the symmetrical single epsilon) this paper would be much better.
- In Table 2, using an epsilon greater than 1 seems to always decrease the error with a seeming greater impact when theta and theta' are close. This trend should be explained. In general, the real-world application was the weakest section. Expounding up on it more, running more revealing experiments (potentially on an actual problem in addition to benchmarks), and providing theoretical motivation would greatly improve the paper.
- In the introduction is is emphasized how this compares to supervised learning but the explanation is how this compares to unsupervised clustering is much more terse. Another sentence or two explaining why using the resulting cluster identifications for binary labeling is inferior to the "arbitrary binary classifier" would help. It's clear in the author's application because one would like to use all data available, including the class priors, for classification.

Minor issues:
-At the bottom of page 3 the authors state, " In fact, these two are fairly different, and the differences are reviewed and discussed in Menon et al. (2015) and van Rooyen & Williamson (2018). " It would be clearer to immediately state the key difference instead of waiting until the end of the paragraph.
- In the first sentence of Section 3.1 "imagining" is mistyped as "imaging."
- What does "classifier-calibrated" mean in Section 3.1?
- In Section 3.1, "That is why by choosing a model G, g∗ = arg ming∈G R(g) is changed as the target to which" was a bit unclear at first. The phrase "is changed as the target to which" was confusing because of the phrasing. Upon second read, the meaning was clear.
- In the introduction it was stated "impossibility is a proof by contradiction, and the possibility is a proof by construction." It would be better to (re)state this with each theorem. I was immediately curious about the proof technique after reading the theorem but no elaboration was provided (other than see the appendix). The footnote with the latter theorem is helpful as it alludes to the kind of construction used without being overly detailed.
- In section 5.2, in the next to last sentence of the first paragraph there are some issues with missing spaces.
- Some more experiment details, e.g. hyperparameter tuning, could be explained in the appendix for reproducibility.

---

> ### Author Response · Authors · 2018-11-20
> **Thank you so much for your time and constructive comments**
>
> We are trying to expand Table 2 by adding noises of different directions to the training class priors (experiments on MNIST are finished but experiments on CIFAR-10 are quite slow; we will update the submission later). Please find our detailed responses below.

---

> > ### Author Response · Authors · 2018-11-20
> > **Meaning of “classification calibrated”**
> >
> > Classification-calibrated loss functions are defined in “P. L. Bartlett, M. I. Jordan, and J. D. McAuliffe. Convexity, classification, and risk bounds. Journal of the American Statistical Association, 2006.” Briefly speaking, using such a surrogate loss guarantees that under mild assumption on the model, the learned model will converge to the Bayes optimal classifier, as the number of training data goes to infinity. Almost all popular losses are classification calibrated---actually, a monotonic and differentiable loss should be classification calibrated, if its gradient at zero is negative.

---

> > ### Author Response · Authors · 2018-11-20
> > **Why using the resulting cluster identifications for binary labeling is inferior to the “arbitrary binary classifier”?**
> >
> > Note that given only U data for training, the most straightforward idea is to use clustering, in particular discriminative clustering which is also known as “unsupervised classification”. This solution is usually suboptimal. A minor reason is that clustering methods are not always compatible with state-of-the-art deep models, but there are two major reasons.
> >
> > First, successful translation of clustering results into classification results exclusively relies on an assumption, namely one cluster exactly corresponds to one class. If we have one cluster formed by a few geometrically close classes, or one class formed by several geometrically separated clusters (as in our experiments), this assumption would be violated and we would fail in translating clusters into meaningful classes. It may happen that clustering results are perfect while classification results are poor.
> >
> > Second, clustering must introduce additional geometric or information-theoretic assumptions (for example, by following the large margin principle and the information-maximization principle). The learning objectives of clustering methods are built upon these additional assumptions. It is very difficult to measure the distance or similarity according to the geometry for complex data in high-dimensional spaces. On the other hand, we employ ERM and rely on the same assumptions of supervised deep learning: the smoothness assumption for supervised learning and the composition-of-factors assumption for deep learning (Section 5.11.2, the DL book). Therefore, we prefer ERM to clustering methods.
> >
> > BTW, the argument was that using clustering is inferior to ERM rather than using clustering is inferior to arbitrary binary classifier. The difference between learning objectives is more critical than the difference between models to be learned.

---

> > ### Author Response · Authors · 2018-11-20
> > **Real-world applications**
> >
> > Two sets of U data with different class priors may be collected from different places or time points. For example, considering morbidity rates, they can be potential patient data collected from urban and rural areas; considering food preferences, they can be potential customer data collected from the Northern and Southern China; likewise, considering approval rates, they can be unlabeled voter data collected in two years.
> >
> > Note that in the seminal paper on learning from label proportions “N. Quadrianto, A. J. Smola, T. S. Caetano, and Q. V. Le. Estimating labels from label proportions. JMLR, 2009”, there are many potential applications in areas like e-commerce, politics, spam filtering and improper content detection. The two problem settings are different yet closely related, and thus those can also be our potential applications.

---

> > ### Author Response · Authors · 2018-11-20
> > **How does the natural extension of UU learning extend beyond the binary setting?**
> >
> > Great question! A similar question has been addressed in a previous reply entitled “Relationship to learning from label proportions (LLP) and natural extensions to k classes with k U sets”. The main message can be summarized as follows. LLP makes use of the mean operator technique for linear-odd losses, and it can naturally handle k classes with k U sets but it cannot learn nonlinear classifiers. The proposed method makes use of the risk rewrite technique from learning with noisy labels, and it can naturally learn nonlinear classifiers but it cannot handle k classes with k U sets.
> >
> > We think the technical difficulty is how to connect the k-class learning problem to learning with noisy labels. For binary classification, this connection is obvious: we regard the U set with larger class prior as the corrupted positive class and the U set with smaller class prior as the corrupted negative class. For multi-class classification with multiple U sets where all class priors are given, we can construct combinatorial many mappings from a U set to a corrupted class, and we lack a measure of the quality of these mappings. This should be the first step for extending this paper beyond binary classification.

---

> > ### Comment · AnonReviewer4 · 2018-11-24
> > **Thank you**
> >
> > Thank you for your many insightful clarifications and expanding your experiments. I look forward to seeing more work in the future!

---

### Official Review · AnonReviewer1 · 2018-11-15
**Interesting paper covering proofs as well as experiments on a newly defined unbiased risk estimator for unlabeled classification**

**Rating:** 8
**Confidence:** 3

**Review:**

This paper proposes a methodology for training any binary classifier from only unlabeled data. They proved that it is impossible to provide an unbiased estimator if having only a single set of unlabeled data, however, they provide an empirical risk minimization method for only two sets of unlabeled data where all the class priors are given. Some experiments and comparisons with state-of-the-art are provided, together with a study on the robustness of the method.

pros:

- The paper is clear, and it provides an interesting proven statement as well as a methodology that can be applied directly. Because they show that only two sets with different (and known) priors are sufficient to have an unbiased estimator, the paper has a clear contribution.
- The impact of the method is a clear asset, because learning from unlabeled data is applicable to a large number of tasks and is raising attention in the last years.
- The large literature on the subject has been well covered in the introduction.
- The importance made on the integration of the method to state-of-the-art classifiers, such as the deep learning framework, is also a very positive point.
- The effort made in the experiments, by testing the performance as well as the robustness of the method with noisy training class priors is very interesting.

remarks:

- part 4.1 : the simplification is interesting. However, the authors say that this simplification is easier to implement in many deep learning frameworks. Why is that?
- part 4.2 : the consistency part is too condensed and not clear enough.
- experiments : what about computation time?
- More generally, I wonder if the authors can find examples of typical problems for classification from unlabeled data with known class priors and with at least two sets?

minor comments:
- part 1: 'but also IN weakly-supervised learning'
- part 2. related work : post- precessing --> post-processing
- part 2. related work : it is proven THAT the minimal number of U sets...
- part 2. related work : In fact, these two are fairly different --> not clear, did you mean 'Actually, ..' ?
- part 4.1 : definition 3. Why naming l- and l+ the corrected loss functions? both of them integrate l(z) and l(-z), so it can be confusing.
- part 5.1 Analysis of moving ... closer: ... is exactly THE same as before.
- part 5.2 : Missing spaces : 'from the webpage of authors.Note ...' and 'USPS datasetsfor the experiment ...'

---

> ### Author Response · Authors · 2018-11-21
> **Thank you so much for your time and constructive comments**
>
> Thanks for pointing out many typos; we will fix them accordingly. Please find our detailed responses below.
>
> Q: Why is the simplification easier to implement in deep learning frameworks?
> A: Sorry for not explaining it. The simplified risk estimator is standard cost-sensitive learning, and thus we can reuse existing codes for cost-sensitive learning, such as importance reweighting by plugging alpha and alpha’ into the codes. However, the original risk estimator needs to be implemented since it is new and cannot be reduced to existing objective functions.
>
> Q: What about computation time in experiments?
> A: Please note that the proposed method just offers a new objective function. After specifying a model, this objective function can be minimized by any optimization algorithm. Specifically, we applied standard SGD for MNIST and Fashion-MNIST and Adam for SVHN and CIFAR-10. Here the proposed method would not add any more computational burden, so that the computation time simply depends on how many epochs we would like to train the model.
>
> Q: Examples of typical problems for classification from two sets of U data with known class priors
> A: Two sets of U data with different class priors may be collected from different places or time points. For example, considering morbidity rates, they can be potential patient data collected from urban and rural areas; considering food preferences, they can be potential customer data collected from the Northern and Southern China; likewise, considering approval rates, they can be unlabeled voter data collected in two years.
>
> Note that in the seminal paper on learning from label proportions “N. Quadrianto, A. J. Smola, T. S. Caetano, and Q. V. Le. Estimating labels from label proportions. JMLR, 2009”, there are many potential applications in areas like e-commerce, politics, spam filtering and improper content detection. The two problem settings are different yet closely related, and thus those can also be our potential applications.
>
> Q: Why naming l- and l+ the corrected loss functions?
> A: We apologize for the confusion. The notation (i.e., l+ and l-) indicates the U set with larger/smaller class prior is regarded as the corrupted P/N dataset. The name is also from learning with noisy labels. Since our training data are corrupted data, using the original loss l means regarding the corrupted data as clean data and will cause learning to be biased and inconsistent. In order to “correct” this effect, the loss has to be corrected so that the corrected loss is perfectly compatible with the corrupted data.

---

> > ### Comment · AnonReviewer1 · 2018-11-29
> > **thank you for the answers**
> >
> > The authors have responded to my questions, and I have no other comment to make.

---

### Official Review · AnonReviewer5 · 2018-11-16
**An interesting contribution and a thorough review of the existing work in the field**

**Rating:** 7
**Confidence:** 4

**Review:**

The authors propose an unbiased estimator that allows for training models with weak supervision on two unlabeled datasets with known class priors. The theoretical properties of the estimator are discussed and an empirical evaluation shows promising performance.

The paper provides a thorough overview of the related work.
The experiments compare to the relevant baselines.

Minor remarks:

The writing seems like it could be improved in multiple places and the main thing that makes some sections of the paper hard to follow is that the concepts often get mentioned and discussed before they are formally defined/introduced. Concepts that are introduced via citations should also be explained even if not in-depth.

Figure 2: the curves suggest that the models should have been left to train for a longer time - some of the small PN and small PN prior-shift risks are still decreasing

Figure 2: the scaling seems inconsistent - the leftmost subplot in each row doesn’t start at (0,0) in the lower left corner, unlike the other subplots in each row - and it should probably be the same throughout - no need to be showing the negative space.

Figure 2: maybe it would be good to plot the different lines in different styles (not just colors) - for BW print and colorblind readers

For small PN and small PN prior-shift, the choice of 10% seems arbitrary. At what percentage do the supervised methods start displaying a clear advantage - for the experiments in the paper?

When looking into the robustness wrt noise in the training class priors, both are multiplied by the same epsilon coefficient. In a more realistic setting the priors might be perturbed independently, potentially even in a different direction. It would be nice to have a more general experiment here, measuring the robustness of the proposed approach in such a way.

5.2 typo: benchmarksand ; datasetsfor

---

> ### Author Response · Authors · 2018-11-21
> **Thank you so much for your time and constructive comments**
>
> And please find our responses below.
>
> Q: For small PN and small PN prior-shift, the choice of 10% seems arbitrary
> A: Yes, using 10% data for training is a bit arbitrary, but it follows the tradition in semi-supervised learning where it is common to give 10% labeled data. Some recent semi-supervised papers give slightly less than 10% labeled data, for example, 4k labeled data for CIFAR-10 in “temporal ensembling” from ICLR 2017, “mean teachers” from NIPS 2017, “smooth neighbors on teacher graphs” from CVPR 2018, and “compact latent space clustering” from ICML 2018. Note that section 5.1 is more a proof of concept and illustration of properties, and hence this arbitrary choice should be a safe choice.
>
> Q: At what percentage do the supervised methods start displaying a clear advantage?
> A: This is a great question but hard to answer. The proposed UU learning is model-independent. However, this doesn’t mean the best model for PN learning is the best model for UU learning due to the memorization in deep networks (“a closer look at memorization in deep networks” from ICML 2017). At what percentage PN learning is clearly better than UU learning mainly depends on 4 factors: first of all, the dataset; second, the values of theta and theta’, which naturally measure how far UU learning is away from PN learning; third, the model capacity in terms of memorizing signals and noises with different speeds---it is conjectured that skip connections themselves have certain regularization effects against label noises; finally, the optimization algorithm, especially the learning rate as a function of the epoch number.
>
> Q: The curves in Fig. 2 suggest that the models should have been trained for longer time
> A: Thanks for the suggestion! We are working on the experiments for extending the number of max epochs for training from 200 to 500; we will see if standard supervised learning with limited data can be significantly better than now.
>
> Q: Try a more realistic setting that the priors are perturbed in a different direction
> A: We have launched new experiments by scaling the training class priors differently. Experiments on MNIST are finished but experiments on CIFAR-10 are quite slow; we will update the submission later. The experimental results on MNIST show that the proposed method still performs reasonably well.

---

### Author Response · Authors · 2018-11-24
**Revision uploaded**

We would like to thank all reviewers for their helpful comments! We have now updated our submission accordingly.
The key modifications of the revised version include:

1. extend the number of max epochs for training from 200 to 500 in the benchmark experiments with deep neural networks (please see Fig. 2),
a table of final test risk is also added (please see Table 4 in Appendix C.2);
2. update Table 2 by adding noises of different directions to the training class priors.

---

### Meta-Review · Area_Chair1 · 2018-12-17
**Nice results for learning a classifier from unlabeled data**

**Confidence:** 5
**Recommendation:** Accept (Poster)

**Metareview:**

This paper studies the task of learning a binary classifier from only unlabeled data. They first provide a negative result, i.e., they show it is impossible to learn an unbiased estimator from a set of unlabeled data. Then they provide an empirical risk minimization method which works when given two sets of unlabeled data, as well as the class priors.

The four submitted reviews were unanimous in their vote to accept. The results are impactful, and might make for an interesting oral presentation.